# Admission systolic blood pressure as a prognostic predictor of acute decompensated heart failure: A report from the KCHF registry

Yuichi Kawase[1], Takao Kato[2]*, Takeshi Morimoto[3], Reo Hata[2], Ryosuke Murai[1], Takeshi Tada[1], Harumi Katoh[1], Kazushige Kadota[1], Erika Yamamoto[2], Hidenori Yaku[2], Yasutaka Inuzuka[4], Yodo Tamaki[5], Neiko Ozasa[2], Yusuke Yoshikawa[2], Moritake Iguchi[6], Kazuya Nagao[7], Yukihito Sato[8], Koichiro Kuwahara[9], Takeshi Kimura[2]

1 Department of Cardiovascular Medicine, Kurashiki Central Hospital, Okayama, Japan, 2 Department of Cardiovascular Medicine, Kyoto University Graduate School of Medicine, Kyoto, Japan, 3 Clinical Epidemiology, Hyogo College of Medicine, Nishinomiya, Japan, 4 Cardiovascular Medicine, Shiga General Hospital, Moriyama, Japan, 5 Department of Cardiology, Tenri Hospital, Nara, Japan, 6 Department of Cardiology, National Hospital Organization Kyoto Medical, Kyoto, Japan, 7 Department of Cardiology, Osaka Red Cross Hospital, Osaka, Japan, 8 Department of Cardiology, Hyogo Prefectural Amagasaki General Medical Center, Amagasaki, Hyogo, Japan, 9 Department of Cardiovascular Medicine, Shinshu University Graduate School of Medicine, Nagano, Japan

* tkato75@kuhp.kyoto-u.ac.jp

**Data Availability Statement:** The minimal data set is ethically restricted by the Institutional Review Board of Kyoto University Hospital. This is because

## Abstract

### Background

Admission systolic blood pressure has emerged as a predictor of postdischarge outcomes of patients with acute decompensated heart failure; however, its validity in varied clinical conditions of this patient subset is unclear. The aim of this study was to further explore the prognostic value of admission systolic blood pressure in patients with acute decompensated heart failure.

### Methods

The Kyoto Congestive Heart Failure (KCHF) registry is a prospective, observational, multi-center cohort study enrolling consecutive patients with acute decompensated heart failure from 19 participating hospitals in Japan. Clinical characteristics at baseline and prognosis were examined by the following value range of admission systolic blood pressure: <100, 100–139, and ≥140 mmHg. The primary outcome measure was defined as all-cause death after discharge. Subgroup analyses were done for prior hospitalization for heart failure, hypertension, left ventricular ejection fraction, and medications at discharge. We excluded patients with acute coronary syndrome or insufficient data.

### Results

We analyzed 3564 patients discharged alive out of 3804 patients hospitalized for acute decompensated heart failure. In the entire cohort, lower admission systolic blood pressure was associated with poor outcomes (1-year cumulative incidence of all-cause death: <100 mmHg, 26.8%; 100–139 mmHg, 20.2%; and ≥140 mmHg, 15.1%, p<0.001). The

the secondary use of the data was to be reviewed by the Ethics Commission at the time of the initial application. Data are available from the Ethics Committee (contact via TK or directly to ethcom@kuhp.kyoto-u.ac.jp) for researchers who meet the criteria for access to confidential data.

**Funding:** This study was supported by grant 18059186 from the Japan Agency for Medical Research and Development (Drs T. Kato, Kuwahara, and Ozasa). The funders had no role in study design, data collection and analysis, decision to publish, or preparation of the manuscript.

**Competing interests:** The authors have declared that no competing interests exist.

magnitude of the effect of lower admission systolic blood pressure for postdischarge all-cause death was greater in patients with prior hospitalization for heart failure, heart failure with reduced left ventricular ejection fraction, and β-blocker use at discharge than in those without.

## Conclusions

Admission systolic blood pressure is useful for postdischarge risk stratification in patients with acute decompensated heart failure. Its magnitude of the effect as a prognostic predictor may differ across clinical conditions of patients.

## Introduction

Acute decompensated heart failure (ADHF) is globally one of the most common causes of hospitalization with high rates of in-hospital and postdischarge mortality and rehospitalization. Low admission systolic blood pressure (SBP) is a well-known prognostic predictor of in-hospital outcomes in ADHF patients. In the Organized Program to Initiate Lifesaving Treatment in Hospitalized Patients with Heart Failure (OPTIMIZE-HF) registry and the Acute Decompensated Heart Failure National Registry (ADHERE), low admission SBP was identified as a predictor of in-hospital mortality in HF patients [1, 2]. In the Finnish Acute Heart Failure (FINN-AKVA) study and the study by Nunez et al, low admission SBP was identified as a predictor of postdischarge mortality in HF patients [3, 4]. Nevertheless, it is unclear whether the validity of low admission SBP in predicting in-hospital and long-term clinical outcomes is consistent across various clinical subtypes of ADHF patients. The aim of this study was to determine whether the impact of admission SBP on long-term prognosis in modern medical care for ADHF tended to be similar to that reported by previous studies by exploring the prognostic value of low admission SBP using the data from a large Japanese observational database of ADHF patients.

## Materials and methods

### Study design and patient population

The Kyoto Congestive Heart Failure (KCHF) registry is a physician-initiated, prospective, observational, multicenter cohort study enrolling consecutive ADHF patients from 19 participating hospitals between October 1, 2014 and March 31, 2016. The 19 hospitals were either secondary or tertiary hospitals, including both rural and urban as well as large and small ones in Japan. One-year clinical follow-up data with an allowance of 1 month were collected in October 2017. The attending physicians or research assistants at each hospital collected clinical event data after the index hospitalization from medical records or from patients, their family members, or their referring physicians under patient consent. Follow-up was commenced on the day of hospital discharge. The details of the KCHF registry have been described previously [5–7]. Briefly, ADHF was defined according to the modified Framingham criteria, and we enrolled consecutive ADHF patients who had undergone HF-specific treatment involving intravenous drug administration within 24 hours after hospital presentation. Patient records were anonymized before analysis.

### Ethics

The investigation conformed to the principles outlined in the Declaration of Helsinki. The study was approved by the institutional review boards of Kyoto University Graduate School of

Medicine (approval number: E2311); Shiga General Hospital (approval number: 20141120–01); Tenri Hospital (approval number: 640); Kobe City Medical Center General Hospital (approval number: 14094); Hyogo Prefectural Amagasaki General Medical Center (approval number: Rinri 26–32); National Hospital Organization Kyoto Medical Center (approval number: 14–080); Mitsubishi Kyoto Hospital (approved 11/12/2014); Okamoto Memorial Hospital (approval number: 201503); Japanese Red Cross Otsu Hospital (approval number: 318); Hikone Municipal Hospital (approval number: 26–17); Japanese Red Cross Osaka Hospital (approval number: 392); Shimabara Hospital (approval number: E2311); Kishiwada City Hospital (approval number: 12); Kansai Electric Power Hospital (approval number: 26–59); Shizuoka General Hospital (approval number: Rin14-11-47); Kurashiki Central Hospital (approval number: 1719); Kokura Memorial Hospital (approval number: 14111202); Kitano Hospital (approval number: P14-11-012); and Japanese Red Cross Wakayama Medical Center (approval number: 328). The study protocol met the conditions of the Japanese Ethical Guidelines for Epidemiological Studies [5–7].

### Definitions and outcomes

SBP measured at the emergency outpatient service was used as admission SBP, which was divided into the following three groups according to the Clinical Scenario classification: <100 mmHg (low admission SBP); 100 to 139 mmHg (intermediate admission SBP); and ≥140 mmHg (high admission SBP) (Fig 1) [8]. HF was classified into the following three categories on the basis of left ventricular ejection fraction (LVEF): LVEF <40% (HF with reduced EF [HFrEF]), LVEF 40% to 49% (HF with mid-range EF [HFmrEF]), and LVEF ≥50% (HF with preserved EF [HFpEF]). Definitions of other baseline factors have been described previously [5–7]. The primary outcome measure was defined as all-cause death after discharge. The

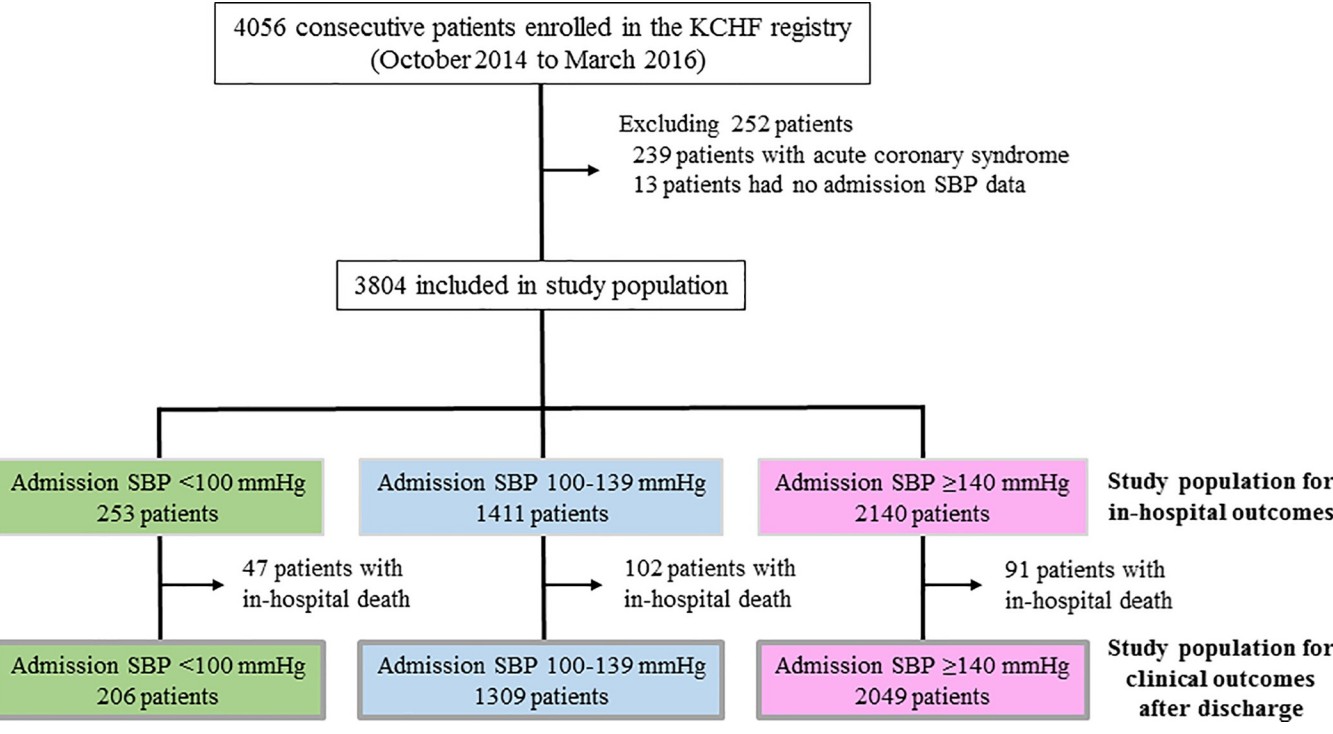

**Fig 1. Study patient flow.** KCHF = Kyoto Congestive Heart Failure, SBP = systolic blood pressure.

secondary outcome measures included in-hospital all-cause death, in-hospital cardiovascular death, in-hospital noncardiovascular death, cardiovascular death after discharge, noncardiovascular death after discharge, and hospitalization for HF. Death was considered cardiovascular in origin unless obvious noncardiovascular causes were identified. Cardiovascular death included death related to HF, death related to stroke, sudden death, and death from other cardiovascular causes. Hospitalization for HF was defined as hospitalization due to worsening HF requiring intravenous drug administration.

## Statistical analysis

The main analysis was the comparison of the primary and secondary outcome measures during hospitalization and after discharge across the three groups based on admission SBP. We also compared patient characteristics on admission, management during hospitalization, and in-hospital outcomes across the same three groups. Subgroup analyses of the association of admission SBP with primary and secondary outcome measures during hospitalization and after discharge were conducted for prior hospitalization for HF, hypertension, and LVEF. Also, those after discharge alone were conducted for β-blocker use, angiotensin converting enzyme inhibitor or angiotensin II receptor blocker use, and calcium channel blocker (CCB) use at discharge.

Categorical variables are presented as numbers and percentages and were compared with the $\chi^2$ test or Fisher's exact test. Continuous variables are presented as the mean and standard deviation or the median and interquartile range. Continuous variables were compared using 1-way analysis of variance or the Kruskal-Wallis test based on their distributions. The Kaplan-Meier method was used to estimate the cumulative incidence of events, and the differences were assessed with the log-rank test.

To estimate the effect of admission SBP on in-hospital mortality, we used a multivariable logistic regression model not accounting for the time to events due to the evaluation of events during the index hospitalization. We included the following 19 clinically relevant risk-adjusting variables into the model: demographical variables (age ≥80 years, sex, and body mass index <22 kg/m$^2$), variables related to heart failure (prior hospitalization for HF, LVEF <40% by echocardiography), variables related to comorbidities (atrial fibrillation or flutter, hypertension, diabetes mellitus, prior myocardial infarction, prior stroke, current smoker, and chronic lung disease), living status (living alone and ambulatory), vital signs at presentation (admission heart rate <60 bpm), laboratory tests on admission (estimated glomerular filtration rate <30 mL/min/1.73 m$^2$, albumin <3.0 g/dL, sodium <135 mmol/L, and anemia) as well as the three groups based on admission SBP (S1 Table). We selected them based on the clinical relevance to prognosis and the mean values of the data to ensure consistency with our previous report [9]. The adjusted risks of the low and intermediate admission SBP groups, respectively, relative to the high SBP group (reference) for the in-hospital clinical outcome measures are expressed as odds ratios and their 95% confidence intervals. We constructed the same multivariable logistic regression models to evaluate the interaction between the subgroup factors and the risk of admission SBP for in-hospital all-cause death and in-hospital cardiovascular death.

The Cox proportional hazard models were used to estimate the unadjusted and adjusted risks of the low and intermediate admission SBP groups, respectively, relative to the high SBP group (reference) for all-cause death, cardiovascular death, noncardiovascular death, and hospitalization for HF, which are expressed as hazard ratios and their 95% confidence intervals. We included the following 21 clinically relevant risk-adjusting variables into the model: demographical variables (age ≥80 years, sex, and body mass index <22 kg/m$^2$), variables related to heart failure (prior hospitalization for HF, LVEF <40% by echocardiography), variables

related to comorbidities (atrial fibrillation or flutter, hypertension, diabetes mellitus, prior myocardial infarction, prior stroke, current smoker, and chronic lung disease), living status (living alone and ambulatory), laboratory tests on admission (estimated glomerular filtration rate <30 mL/min/1.73 m$^2$, albumin <3.0 g/dL, sodium <135 mmol/L, and anemia), and medications at discharge (angiotensin converting enzyme inhibitors or angiotensin II receptor blockers, and β-blockers) as well as the three groups based on admission SBP. We selected them on the basis of the clinical relevance to prognosis and the mean values of the data to ensure consistency with our previous report [7]. Continuous variables were dichotomized by median values or clinically meaningful reference values. We constructed the same Cox proportional hazard models to evaluate the interaction between the subgroup factors and the risks of the low and intermediate admission SBP groups, respectively, relative to the high SBP group (reference) for the postdischarge clinical outcomes.

All statistical analyses were conducted by two physicians (Y.K. and H.Y.) with JMP 10.0.2 (SAS institute, Cary, NC) or SAS 9.4 (SAS institute). All reported p values were 2-tailed, and <0.05 was considered statistically significant.

## Data sharing

The minimal data set is ethically restricted by the Institutional Review Board of Kyoto University Hospital. This is because the secondary use of the data was to be reviewed by the Ethics Commission at the time of the initial application. Data are available from the Ethics Committee (contact via TK or directly to ethcom@kuhp.kyoto-u.ac.jp) for researchers who meet the criteria for access to confidential data.

## Results

### Study population

Among 4056 patients enrolled in the KCHF registry, the study population for in-hospital outcomes included 3804 patients after excluding 239 patients with acute coronary syndrome and 13 patients with missing data on admission SBP (**Fig 1**). The study population for postdischarge clinical outcomes included 3785 patients who had been discharged alive from the index hospitalization (**Fig 1**).

### Patient characteristics on admission, and management during hospitalization

Regarding the patient characteristics on admission, patients in the low admission SBP group more often had prior hospitalization for HF, cardiomyopathy, and lower LVEF than those in the other two groups. Patients in the high admission SBP group more often had hypertensive heart disease and hypertension than those in the other two groups (**S1 Table**). Regarding the in-hospital management, patients in the high admission SBP group more often used noninvasive positive pressure ventilation and vasodilators, while those in the low admission SBP group more often used inotropes and intra-aortic balloon pumping (**S2 Table**).

### In-hospital outcomes

The rates of all-cause death and cardiovascular death were significantly higher in the low admission SBP group than in the intermediate and high admission SBP groups (all-cause death: 19%, 7%, and 4%; and cardiovascular death: 16%, 5%, and 3%, respectively) (**Table 1**). After adjusting confounders, the excess risks of low and intermediate admission SBP, respectively, relative to high admission SBP were significant for all-cause death and cardiovascular

**Table 1. In-hospital mortality by admission SBP.**

|  | Event | Unadjusted OR | 95% CI | P value | Adjusted OR | 95% CI | P value |
|---|---|---|---|---|---|---|---|
| **All-cause death** |  |  |  |  |  |  |  |
| SBP ≥140 mmHg | 91/2140 (4.3) | 1 (reference) |  |  | 1 (reference) |  |  |
| SBP 100–139 mmHg | 102/1411 (7.2) | 1.75 | 1.31–2.35 | <0.001 | 1.59 | 1.11–2.28 | 0.01 |
| SBP <100 mmHg | 47/253 (19) | 5.14 | 3.49–7.48 | <0.001 | 3.61 | 2.16–5.94 | <0.001 |
| **Cardiovascular death** |  |  |  |  |  |  |  |
| SBP ≥140 mmHg | 62/2140 (2.9) | 1 (reference) |  |  | 1 (reference) |  |  |
| SBP 100–139 mmHg | 72/1411 (5.1) | 1.80 | 1.28–2.55 | <0.001 | 1.72 | 1.12–2.66 | 0.01 |
| SBP <100 mmHg | 40/253 (16) | 6.29 | 4.10–9.56 | <0.001 | 4.25 | 2.37–7.51 | <0.001 |
| **Noncardiovascular death** |  |  |  |  |  |  |  |
| SBP ≥140 mmHg | 29/2140 (1.4) | 1 (reference) |  |  | 1 (reference) |  |  |
| SBP 100–139 mmHg | 30/1411 (2.1) | 1.58 | 0.94–2.66 | 0.08 | 1.24 | 0.67–2.31 | 0.49 |
| SBP <100 mmHg | 7/253 (2.8) | 2.07 | 0.83–4.52 | 0.11 | 1.80 | 0.66–4.44 | 0.24 |

Values are number (%).

SBP = systolic blood pressure, OR = odds ratio, and CI = confidence interval.

death, indicating an incrementally higher risk with lower admission SBP (**Table 1**). The rate of ventricular tachycardia or fibrillation was significantly higher in the low admission SBP group than in the intermediate and high admission SBP groups (11%, 5%, and 3.6%, respectively) (**S3 Table**). The rate of worsening renal function was significantly lower in the low admission SBP group than in the intermediate and high admission SBP groups (19%, 29%, and 40%, respectively) (**S3 Table**). In the subgroup analyses, there was significant interaction of all-cause death during hospitalization with a history of hypertension; the magnitude of the effect of low admission SBP was greater in patients with hypertension than in those without (**Fig 2**). There was no significant interaction of cardiovascular death during hospitalization with all subgroup factors (**Fig 2**).

## Patient characteristics at discharge

Patient characteristics in the study population for postdischarge clinical outcomes were generally consistent with those in the study population for in-hospital outcomes (**Table 2**).

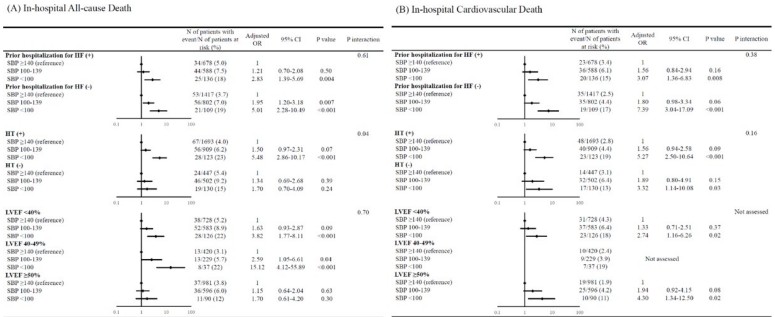

**Fig 2. Subgroup analyses for the effect of admission SBP on in-hospital clinical events.** (A) All-cause death, (B) Cardiovascular death during hospitalization. In the analysis of the effect of admission SBP on in-hospital cardiovascular death, patients with LVEF 40%–49% were not assessed because the number of patients was too small. SBP = systolic blood pressure, OR = odds ratio, CI = confidence interval, HF = heart failure, HT = hypertension, and LVEF = left ventricular ejection fraction.

**Table 2. Patient characteristics at discharge.**

| Variables | Entire cohort (N = 3564) | Admission SBP <100 mmHg (N = 206) | Admission SBP 100–139 mmHg (N = 1309) | Admission SBP ≥140 mmHg (N = 2049) | P value |
|---|---|---|---|---|---|
| Age, years | 77.8 ± 12.0 | 74.3 ± 14.5 | 77.6 ± 12.1 | 78.3 ± 11.7 | 0.001 |
| ≥80 years [a] | 1874 (53) | 88 (43) | 690 (53) | 1096 (53) | 0.01 |
| Men [a] | 1948 (55) | 122 (59) | 747 (57) | 1079 (53) | 0.02 |
| Body mass index at discharge, kg/m² [b] | 21.3 ± 4.2 | 20.8 ± 4.3 | 21.1 ± 4.2 | 21.5 ± 4.2 | 0.003 |
| <22 kg/m² [a] | 2082 (62) | 134 (68) | 795 (63) | 1153 (60) | 0.03 |
| Prior hospitalization for heart failure [a] | 1299 (37) | 111 (56) | 544 (42) | 644 (32) | <0.001 |
| Current smoker [a] | 417 (12) | 15 (7.5) | 131 (10) | 271 (13) | 0.003 |
| Ambulatory at discharge [a] | 2575 (74) | 151 (75) | 936 (72) | 1488 (74) | 0.50 |
| Living alone [a] | 771 (22) | 33 (16) | 277 (21) | 461 (23) | 0.09 |
| **Etiology** | | | | | <0.001 |
| Coronary artery disease | 1014 (28) | 54 (26) | 376 (29) | 584 (29) | |
| Cardiomyopathy | 564 (16) | 72 (35) | 263 (20) | 229 (11) | |
| Hypertensive heart disease | 945 (27) | 11 (5.3) | 187 (14) | 747 (36) | |
| Valvular heart disease | 746 (21) | 41 (20) | 341 (26) | 364 (18) | |
| Others | 295 (8) | 28 (14) | 142 (11) | 125 (6) | |
| **Concomitant diseases** | | | | | |
| Hypertension [a] | 2574 (72) | 95 (46) | 853 (65) | 1626 (79) | <0.001 |
| Diabetes [a] | 1307 (37) | 66 (32) | 452 (35) | 789 (39) | 0.02 |
| Prior myocardial infarction [a] | 791 (22) | 43 (21) | 281 (21) | 467 (23) | 0.60 |
| Prior stroke [a] | 572 (16) | 32 (16) | 210 (16) | 330 (16) | 0.98 |
| Atrial fibrillation or flutter [a] | 1543 (43) | 110 (53) | 679 (52) | 754 (37) | <0.001 |
| Ventricular tachycardia or fibrillation | 150 (4.2) | 29 (14) | 69 (5.3) | 52 (2.5) | <0.001 |
| Malignant neoplasm | 512 (14) | 26 (13) | 166 (13) | 320 (16) | 0.047 |
| Chronic lung disease [a] | 478 (13) | 21 (10) | 176 (13) | 281 (14) | 0.37 |
| Prior percutaneous coronary intervention | 774 (22) | 39 (19) | 272 (21) | 463 (23) | 0.28 |
| Prior coronary artery bypass grafting | 268 (7.5) | 18 (8.7) | 106 (8.1) | 144 (7.0) | 0.41 |
| **Hemodynamic data at discharge** | | | | | |
| Heart rate, bpm | 71 ± 13 | 75 ± 14 | 72 ± 14 | 70 ± 12 | <0.001 |
| Systolic blood pressure, mmHg | 116 ± 18 | 101 ± 16 | 111 ± 16 | 121 ± 18 | <0.001 |
| Diastolic blood pressure, mmHg | 64 ± 12 | 59 ± 12 | 63 ± 12 | 65 ± 13 | <0.001 |
| **Symptoms at discharge** | | | | | |
| NYHA class 3 or 4 | 217 (6.2) | 18 (8.7) | 106 (8.1) | 93 (4.7) | <0.001 |
| Orthopnea | 131 (3.8) | 6 (3.0) | 48 (3.8) | 77 (3.9) | 0.82 |
| Rales | 177 (5.2) | 15 (7.5) | 72 (5.8) | 90 (4.6) | 0.10 |
| Dyspnea on exertion | 931 (27) | 66 (33) | 412 (33) | 453 (23) | <0.001 |
| Jugular venous distention | 227 (6.7) | 23 (12) | 92 (7.3) | 112 (5.7) | 0.003 |
| Peripheral edema | 438 (13) | 31 (16) | 190 (15) | 217 (11) | 0.002 |
| **Chest radiograph at discharge** | | | | | |
| Pulmonary congestion | 295 (8.5) | 23 (11) | 118 (9.2) | 154 (7.7) | 0.10 |
| Pleural effusion | 556 (16) | 43 (21) | 217 (17) | 296 (15) | 0.03 |
| **Laboratory values at discharge** | | | | | |
| Hemoglobin, mg/dL | 11.5 ± 2.2 | 11.3 ± 2.0 | 11.6 ± 2.2 | 11.4 ± 2.2 | 0.03 |
| Anemia [a, c] | 2433 (70) | 141 (71) | 887 (69) | 1405 (71) | 0.59 |
| Serum creatinine, mg/dL | 1.12 (0.86–1.59) | 1.11 (0.89–1.54) | 1.10 (0.83–1.50) | 1.14 (0.87–1.67) | <0.001 |
| Estimated glomerular filtration rate, mL/min/1.73 m² | 43.1 (29.1–58.7) | 43.3 (32.4–59.8) | 44.3 (31.2–61.4) | 41.6 (27.1–56.2) | <0.001 |
| <30 mL/min/1.73 m² [a] | 913 (26) | 48 (24) | 291 (12) | 574 (29) | <0.001 |
| Albumin, g/dL | 3.4 ± 0.5 | 3.3 ± 0.5 | 3.4 ± 0.5 | 3.3 ± 0.5 | 0.09 |

(*Continued*)

**Table 2.** (Continued)

| Variables | Entire cohort (N = 3564) | Admission SBP <100 mmHg (N = 206) | Admission SBP 100–139 mmHg (N = 1309) | Admission SBP ≥140 mmHg (N = 2049) | P value |
|---|---|---|---|---|---|
| <3 g/dL [a] | 619 (20) | 40 (23) | 216 (19) | 363 (20) | 0.38 |
| Serum sodium, mmol/L | 138 ± 3.8 | 138 ± 4.1 | 138 ± 3.9 | 139 ± 3.6 | <0.001 |
| <135 mmol/L [a] | 446 (13) | 42 (21) | 185 (14) | 219 (11) | <0.001 |
| Serum potassium, mmol/L | 4.2 ± 0.5 | 4.2 ± 0.5 | 4.2 ± 0.5 | 4. 2 ± 0.5 | 0.22 |
| Total bilirubin, mg/dL | 0.6 (0.4–0.8) | 0.7 (0.5–1.0) | 0.6 (0.5–0.9) | 0.5 (0.4–0.7) | <0.001 |
| Brain-type natriuretic peptide, pg/mL | 261 (133–510) | 472 (204–793) | 290 (155–539) | 235 (119–454) | <0.001 |
| **Echocardiographic parameters** | | | | | |
| Left ventricular ejection fraction, % | 46 (33–60) | 42 (24–58) | 44 (31–60) | 48 (36–60) | <0.001 |
| <40% [a] | 1319 (37) | 98 (48) | 531 (41) | 690 (34) | |
| 40%–49% | 652 (18) | 29 (14) | 216 (17) | 407 (20) | |
| ≥50% | 1583 (45) | 79 (38) | 560 (43) | 944 (46) | |
| Moderate–severe mitral regurgitation | 1140 (34) | 74 (38) | 501 (40) | 565 (29) | <0.001 |
| Moderate–severe aortic stenosis | 218 (6.5) | 11 (5.7) | 91 (7.4) | 116 (6.0) | 0.29 |
| **Oral medications at discharge** | | | | | |
| β-blocker [a] | 2338 (66) | 143 (69) | 845 (65) | 1350 (66) | 0.36 |
| Mineralocorticoid receptor antagonist [a] | 1615 (45) | 115 (56) | 618 (47) | 882 (43) | <0.001 |
| ACEI or ARB [a] | 2040 (57) | 78 (38) | 662 (51) | 1300 (63) | <0.001 |
| Loop diuretics | 2913 (82) | 170 (83) | 1109 (85) | 1634 (80) | 0.001 |
| Thiazide | 213 (6.0) | 15 (7.3) | 81 (6.2) | 117 (5.7) | 0.61 |
| Tolvaptan | 385 (11) | 52 (25) | 154 (12) | 179 (8.7) | <0.001 |
| Calcium channel blocker | 1241 (35) | 34 (17) | 324 (25) | 883 (43) | <0.001 |

Values are number (%), mean ± standard deviation, or median (interquartile range).

[a] Risk-adjusting variables selected in the Cox proportional hazard models for all-cause death, cardiovascular death, noncardiovascular death, and hospitalization for heart failure.

[b] Body mass index was calculated as weight in kilograms divided by height in meters squared.

[c] Anemia was defined according to the World Health Organization criteria (hemoglobin <12.0 g/dL in women and <13.0 g/dL in men).

SBP = systolic blood pressure, NYHA = New York Heart Association, ACEI = angiotensin-converting enzyme inhibitor, and ARB = angiotensin II receptor blocker.

## Post-discharge clinical outcomes

The median follow-up period was 459 (interquartile range: 352–631) days after discharge, and the 1-year follow-up rate was 94%. The cumulative 1-year incidences of all-cause death and cardiovascular death were incrementally higher with lower admission SBP (**Fig 3A and 3B**). The cumulative 1-year incidence of noncardiovascular death was not different across the three groups (**Fig 3C**). The cumulative 1-year incidence of hospitalization for HF was also significantly higher in the low admission SBP group than in the other two groups (**Fig 3D**). After adjusting for confounders, the excess risk of low admission SBP relative to high SBP remained significant for all-cause death, cardiovascular death, and hospitalization for HF; that of intermediate admission SBP relative to high SBP was also significant for all-cause death and cardiovascular death, but with smaller effect sizes than those of low admission SBP (**Table 3**).

## Subgroup analysis of postdischarge outcomes

In the subgroup analyses of all-cause death, there was significant interaction with certain subgroup factors such as prior hospitalization for HF, LVEF, and β-blocker use at discharge; the

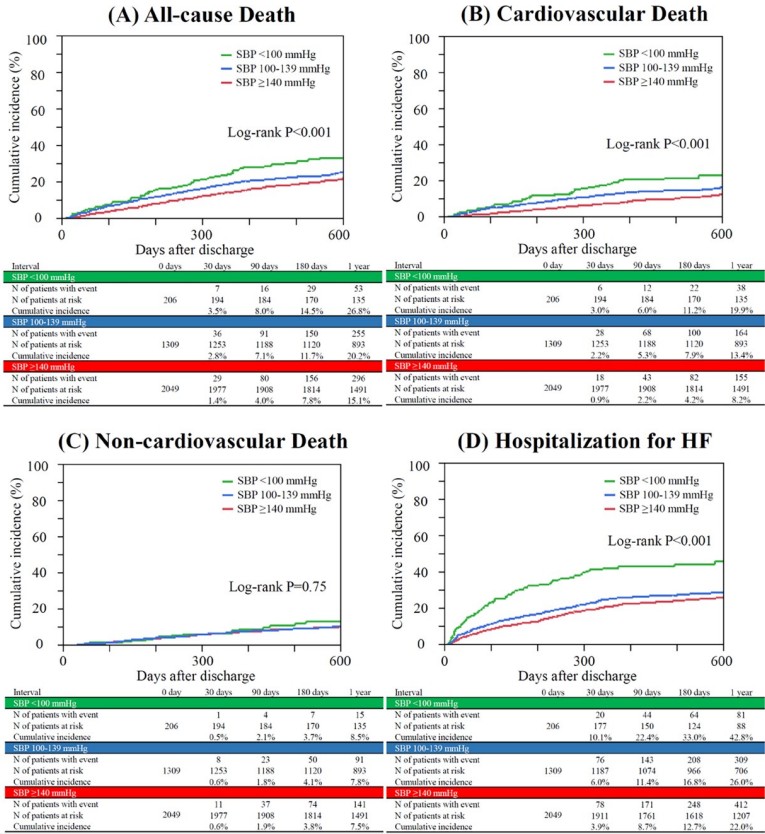

**Fig 3. Kaplan-Meier curves for postdischarge clinical events based on admission SBP status.** (A) All-cause death, (B) Cardiovascular death, (C) Noncardiovascular death, and (D) Hospitalization for HF. Follow-up was commenced on the day of discharge. SBP = systolic blood pressure, HF = heart failure.

magnitude of the effect of lower admission SBP on all-cause death was greater in patients with prior hospitalization for HF, HFrEF, and β-blocker use than in those without (**Fig 4**). In the subgroup analyses of cardiovascular death, there was significant interaction with certain subgroup factors such as prior hospitalization for HF and LVEF; the magnitude of the effect of lower admission SBP on cardiovascular death was greater in patients with prior hospitalization for HF and HFrEF than in those without (**Fig 5**). There was no significant interaction of noncardiovascular death with all subgroup factors (**Fig 6**). In the subgroup analyses of hospitalization for HF, there was significant interaction with certain subgroup factors such as prior hospitalization for HF, hypertension, LVEF, and CCB use; the magnitude of the effect of lower admission SBP on hospitalization for HF was greater in patients with prior hospitalization for HF, no hypertension, HFrEF, and no CCB use than in those without (**Fig 7**).

## Discussion

The findings of this study are as follows: (1) In the entire cohort, lower admission SBP was associated with higher risk of in-hospital and postdischarge all-cause and cardiovascular death and hospitalization for HF, but not with in-hospital and postdischarge noncardiovascular death; (2) The association of low admission SBP with higher postdischarge mortality was greater in patients with previous hospitalization, HFrEF, and β-blocker use; and (3) The association of lower admission SBP with hospitalization for HF was greater in patients without a history of hypertension and with CCB use.

**Table 3. Post-discharge clinical outcomes by admission SBP.**

| | N of patients with events/N of patients at risk (Cumulative 1-year incidence) | Unadjusted HR | 95% CI | P value | Adjusted HR | 95% CI | P value |
|---|---|---|---|---|---|---|---|
| **All-cause death** | | | | | | | |
| SBP ≥140 mmHg | 424/2049 (15) | 1 (reference) | | | 1 (reference) | | |
| SBP 100–140 mmHg | 326/1309 (20) | 1.26 | 1.09–1.45 | 0.002 | 1.26 | 1.06–1.49 | 0.01 |
| SBP <100 mmHg | 67/206 (27) | 1.74 | 1.33–2.23 | <0.001 | 1.64 | 1.21–2.20 | 0.002 |
| **Hospitalization for heart failure** | | | | | | | |
| SBP ≥140 mmHg | 489/2049 (22) | 1 (reference) | | | 1 (reference) | | |
| SBP 100–140 mmHg | 343/1309 (26) | 1.16 | 1.01–1.33 | 0.04 | 1.11 | 0.95–1.30 | 0.20 |
| SBP <100 mmHg | 85/206 (43) | 2.17 | 1.71–2.72 | <0.001 | 1.91 | 1.44–2.50 | <0.001 |
| **Cardiovascular death** | | | | | | | |
| SBP ≥140 mmHg | 229/2049 (8) | 1 (reference) | | | 1 (reference) | | |
| SBP 100–140 mmHg | 208/1309 (13) | 1.49 | 1.23–1.79 | <0.001 | 1.43 | 1.14–1.79 | 0.002 |
| SBP <100 mmHg | 46/206 (20) | 2.21 | 1.59–3.00 | <0.001 | 2.01 | 1.37–2.88 | <0.001 |
| **Noncardiovascular death** | | | | | | | |
| SBP ≥140 mmHg | 195/2049 (8) | 1 (reference) | | | 1 (reference) | | |
| SBP 100–140 mmHg | 118/1309 (8) | 0.99 | 0.79–1.24 | 0.93 | 1.04 | 0.79–1.36 | 0.78 |
| SBP <100 mmHg | 21/206 (9) | 1.18 | 0.73–1.80 | 0.48 | 1.10 | 0.63–1.82 | 0.73 |

The number of patients with events was counted through the entire follow-up period, while the cumulative incidence was truncated at 1 year.

SBP = systolic blood pressure, HR = hazard ratio, and CI = confidence interval.

In this study, admission SBP of ADHF patients was divided into three groups according to the Clinical Scenario classification. As a sensitivity analysis, the Kaplan-Meier method was used to estimate the cumulative incidence of events, and the differences were assessed with the log-rank test using further subdivided range of blood pressure (<100, 100–119, 120–139, 140–159, ≥160 mmHg), and the results were consistent with those of the main analysis (**S1 Fig**).

In this study, lower admission SBP was associated with poor in-hospital and 1-year outcomes, which is in line with previous studies. In the OPTIMIZE-HF registry, low admission SBP was identified as a predictor of short-term mortality in HF patients despite medical therapy [1]. In the FINN-AKVA study, the 1-year mortality rate was higher in HF patients with lower admission SBP [3]. In this large-scale comprehensive registry for ADHF, we also showed the incremental effects of lower SBP on all-cause death, cardiovascular death, and hospitalization for HF.

There are a few plausible mechanisms of the inverse association between admission SBP and poor prognosis of HF patients. Blood pressure is determined by cardiac output and systemic resistance. In ADHF settings, the prompt adaptation of cardiac output and elasticity of arteries and vascular bed was decompensated [10]. In HFrEF patients, low cardiac output was related to low admission SBP; thus, low admission SBP was related to both high in-hospital mortality and poor postdischarge outcomes. In contrast, when cardiac output is normal or slightly reduced, a hypertensive response is expected, particularly in hypertensive patients, as a

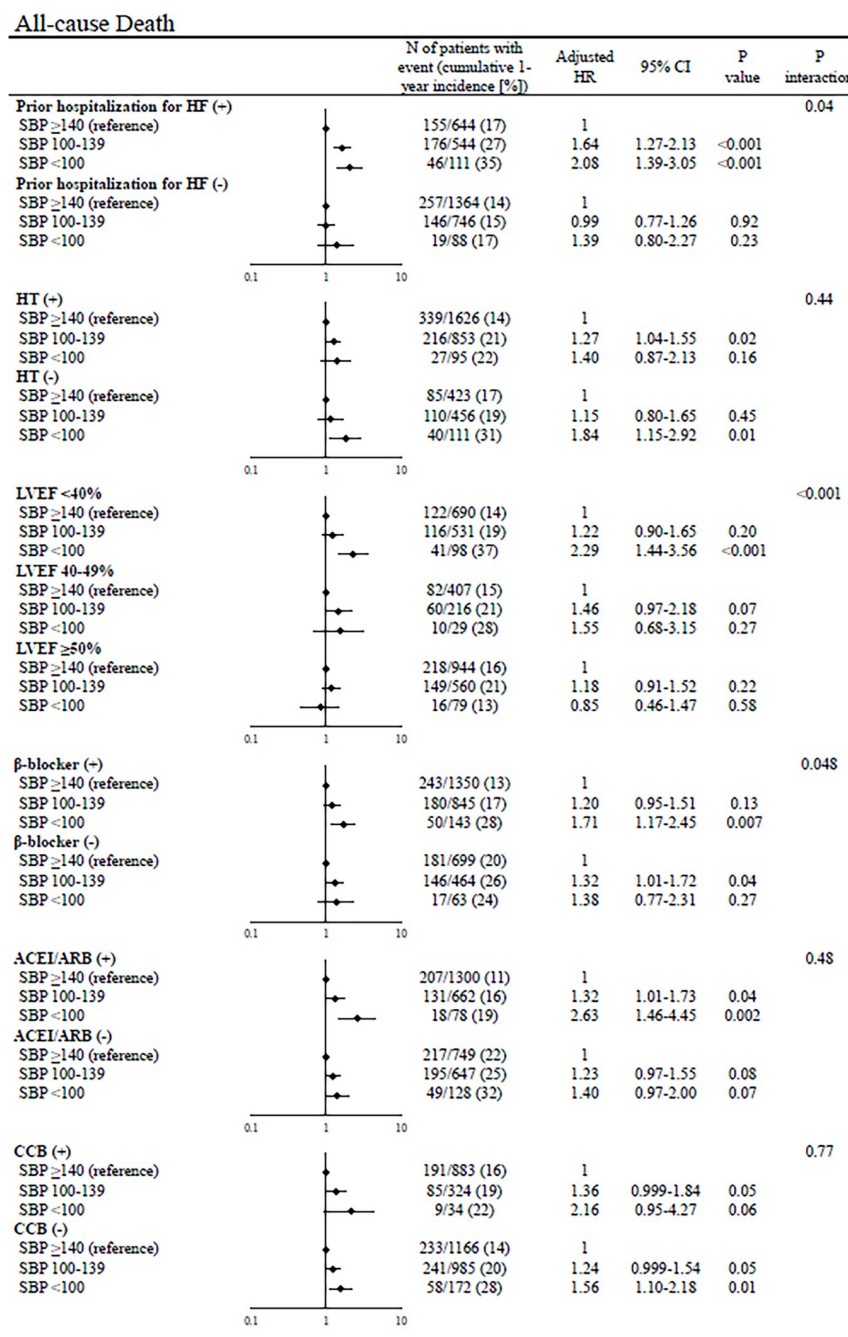

All-cause Death

| | N of patients with event (cumulative 1-year incidence [%]) | Adjusted HR | 95% CI | P value | P interaction |
|---|---|---|---|---|---|
| Prior hospitalization for HF (+) | | | | | 0.04 |
| SBP ≥140 (reference) | 155/644 (17) | 1 | | | |
| SBP 100-139 | 176/544 (27) | 1.64 | 1.27-2.13 | <0.001 | |
| SBP <100 | 46/111 (35) | 2.08 | 1.39-3.05 | <0.001 | |
| Prior hospitalization for HF (-) | | | | | |
| SBP ≥140 (reference) | 257/1364 (14) | 1 | | | |
| SBP 100-139 | 146/746 (15) | 0.99 | 0.77-1.26 | 0.92 | |
| SBP <100 | 19/88 (17) | 1.39 | 0.80-2.27 | 0.23 | |
| HT (+) | | | | | 0.44 |
| SBP ≥140 (reference) | 339/1626 (14) | 1 | | | |
| SBP 100-139 | 216/853 (21) | 1.27 | 1.04-1.55 | 0.02 | |
| SBP <100 | 27/95 (22) | 1.40 | 0.87-2.13 | 0.16 | |
| HT (-) | | | | | |
| SBP ≥140 (reference) | 85/423 (17) | 1 | | | |
| SBP 100-139 | 110/456 (19) | 1.15 | 0.80-1.65 | 0.45 | |
| SBP <100 | 40/111 (31) | 1.84 | 1.15-2.92 | 0.01 | |
| LVEF <40% | | | | | <0.001 |
| SBP ≥140 (reference) | 122/690 (14) | 1 | | | |
| SBP 100-139 | 116/531 (19) | 1.22 | 0.90-1.65 | 0.20 | |
| SBP <100 | 41/98 (37) | 2.29 | 1.44-3.56 | <0.001 | |
| LVEF 40-49% | | | | | |
| SBP ≥140 (reference) | 82/407 (15) | 1 | | | |
| SBP 100-139 | 60/216 (21) | 1.46 | 0.97-2.18 | 0.07 | |
| SBP <100 | 10/29 (28) | 1.55 | 0.68-3.15 | 0.27 | |
| LVEF ≥50% | | | | | |
| SBP ≥140 (reference) | 218/944 (16) | 1 | | | |
| SBP 100-139 | 149/560 (21) | 1.18 | 0.91-1.52 | 0.22 | |
| SBP <100 | 16/79 (13) | 0.85 | 0.46-1.47 | 0.58 | |
| β-blocker (+) | | | | | 0.048 |
| SBP ≥140 (reference) | 243/1350 (13) | 1 | | | |
| SBP 100-139 | 180/845 (17) | 1.20 | 0.95-1.51 | 0.13 | |
| SBP <100 | 50/143 (28) | 1.71 | 1.17-2.45 | 0.007 | |
| β-blocker (-) | | | | | |
| SBP ≥140 (reference) | 181/699 (20) | 1 | | | |
| SBP 100-139 | 146/464 (26) | 1.32 | 1.01-1.72 | 0.04 | |
| SBP <100 | 17/63 (24) | 1.38 | 0.77-2.31 | 0.27 | |
| ACEI/ARB (+) | | | | | 0.48 |
| SBP ≥140 (reference) | 207/1300 (11) | 1 | | | |
| SBP 100-139 | 131/662 (16) | 1.32 | 1.01-1.73 | 0.04 | |
| SBP <100 | 18/78 (19) | 2.63 | 1.46-4.45 | 0.002 | |
| ACEI/ARB (-) | | | | | |
| SBP ≥140 (reference) | 217/749 (22) | 1 | | | |
| SBP 100-139 | 195/647 (25) | 1.23 | 0.97-1.55 | 0.08 | |
| SBP <100 | 49/128 (32) | 1.40 | 0.97-2.00 | 0.07 | |
| CCB (+) | | | | | 0.77 |
| SBP ≥140 (reference) | 191/883 (16) | 1 | | | |
| SBP 100-139 | 85/324 (19) | 1.36 | 0.999-1.84 | 0.05 | |
| SBP <100 | 9/34 (22) | 2.16 | 0.95-4.27 | 0.06 | |
| CCB (-) | | | | | |
| SBP ≥140 (reference) | 233/1166 (14) | 1 | | | |
| SBP 100-139 | 241/985 (20) | 1.24 | 0.999-1.54 | 0.05 | |
| SBP <100 | 58/172 (28) | 1.56 | 1.10-2.18 | 0.01 | |

**Fig 4. Subgroup analyses of the effects of admission SBP on all-cause death.** The number of patients with events was counted through the entire follow-up period, while the cumulative incidence was truncated at 1 year.
SBP = systolic blood pressure, HR = hazard ratio, CI = confidence interval, HF = heart failure, LVEF = left ventricular ejection fraction, ACEI = angiotensin-converting enzyme inhibitor, ARB = angiotensin II receptor blocker, and CCB = calcium channel blocker.

result of sympathetic and neurohormonal activation. Thus, at the time of discharge when they were under drug therapy, admission SBP may have had smaller effects on HFmrEF and HFpEF patients. In this study, we examined whether admission SBP was a predictor of long-term prognosis, assuming that the response of blood pressure during acute exacerbation of HF

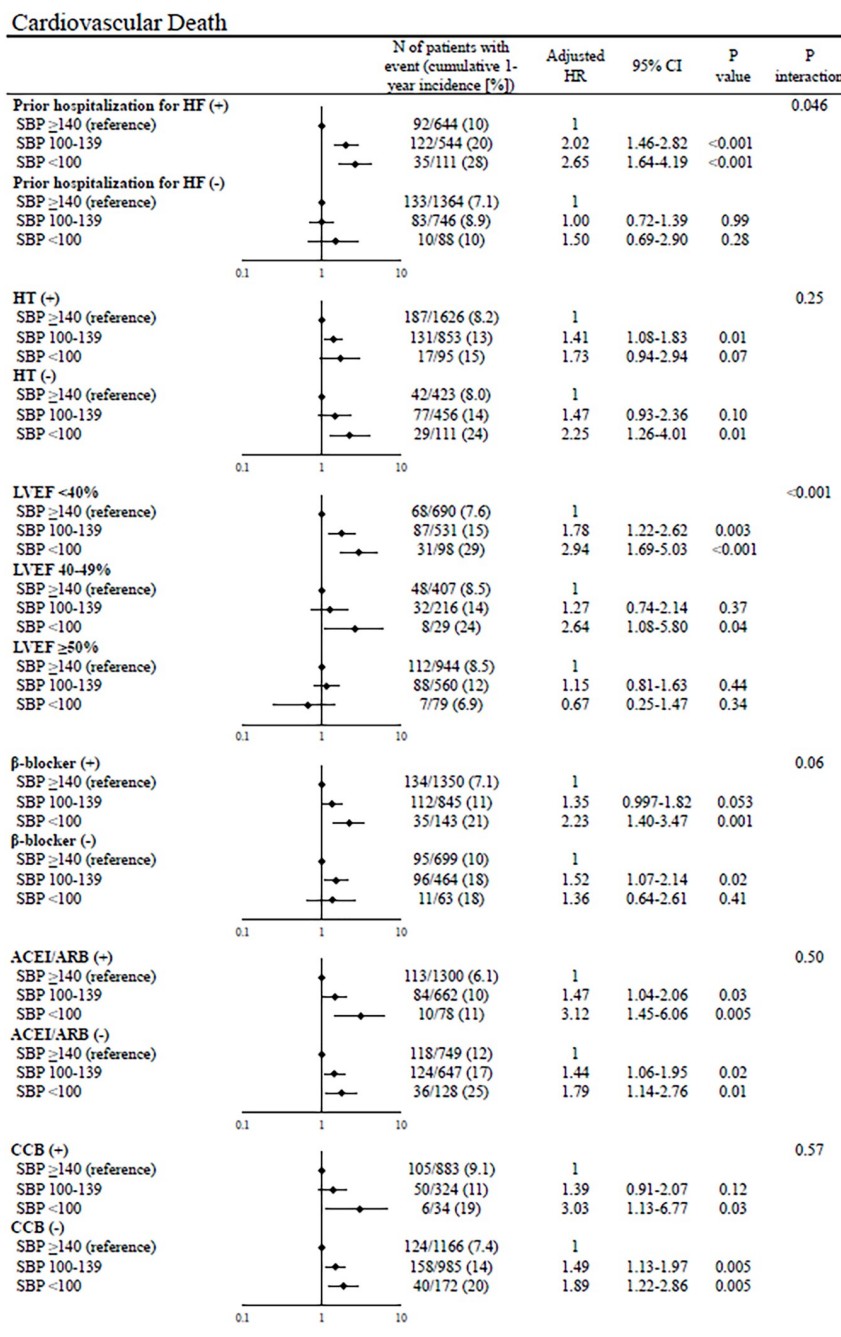

**Fig 5. Subgroup analyses of the effects of admission SBP on cardiovascular death.** The number of patients with events was counted through the entire follow-up period, while the cumulative incidence was truncated at 1 year. SBP = systolic blood pressure, HR = hazard ratio, CI = confidence interval, HF = heart failure, LVEF = left ventricular ejection fraction, ACEI = angiotensin-converting enzyme inhibitor, ARB = angiotensin II receptor blocker, and CCB = calcium channel blocker.

was due to the mechanism described above. These findings are supported by the observation that the risk of all-cause death and cardiovascular death in the low admission SBP group was greater in patients with prior HF hospitalization and β-blocker use when considering the high rate of repeated hospitalization and prescription of β-blocker to HFrEF patients. Also, various

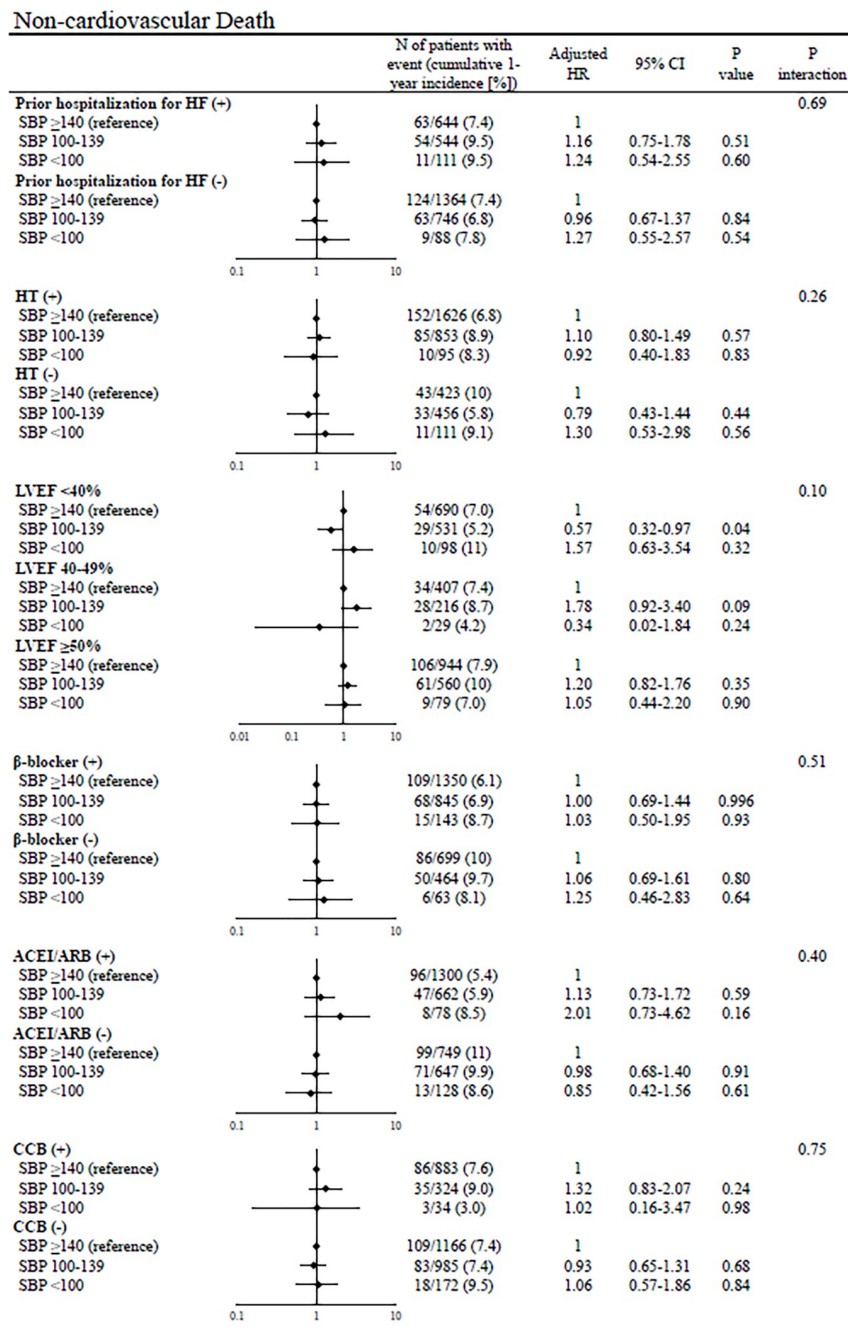

**Fig 6. Subgroup analyses of the effects of admission SBP on noncardiovascular death.** The number of patients with events was counted through the entire follow-up period, while the cumulative incidence was truncated at 1 year. SBP = systolic blood pressure, HR = hazard ratio, CI = confidence interval, HF = heart failure, LVEF = left ventricular ejection fraction, ACEI = angiotensin-converting enzyme inhibitor, ARB = angiotensin II receptor blocker, and CCB = calcium channel blocker.

causes of death in the EF-based classification groups might have contributed to the above outcomes. Our previous study demonstrated that the occurrence rate of noncardiovascular death was higher in HFmrEF and HFpEF patients than in HFrEF patients [11].

## Hospitalization for Heart Failure

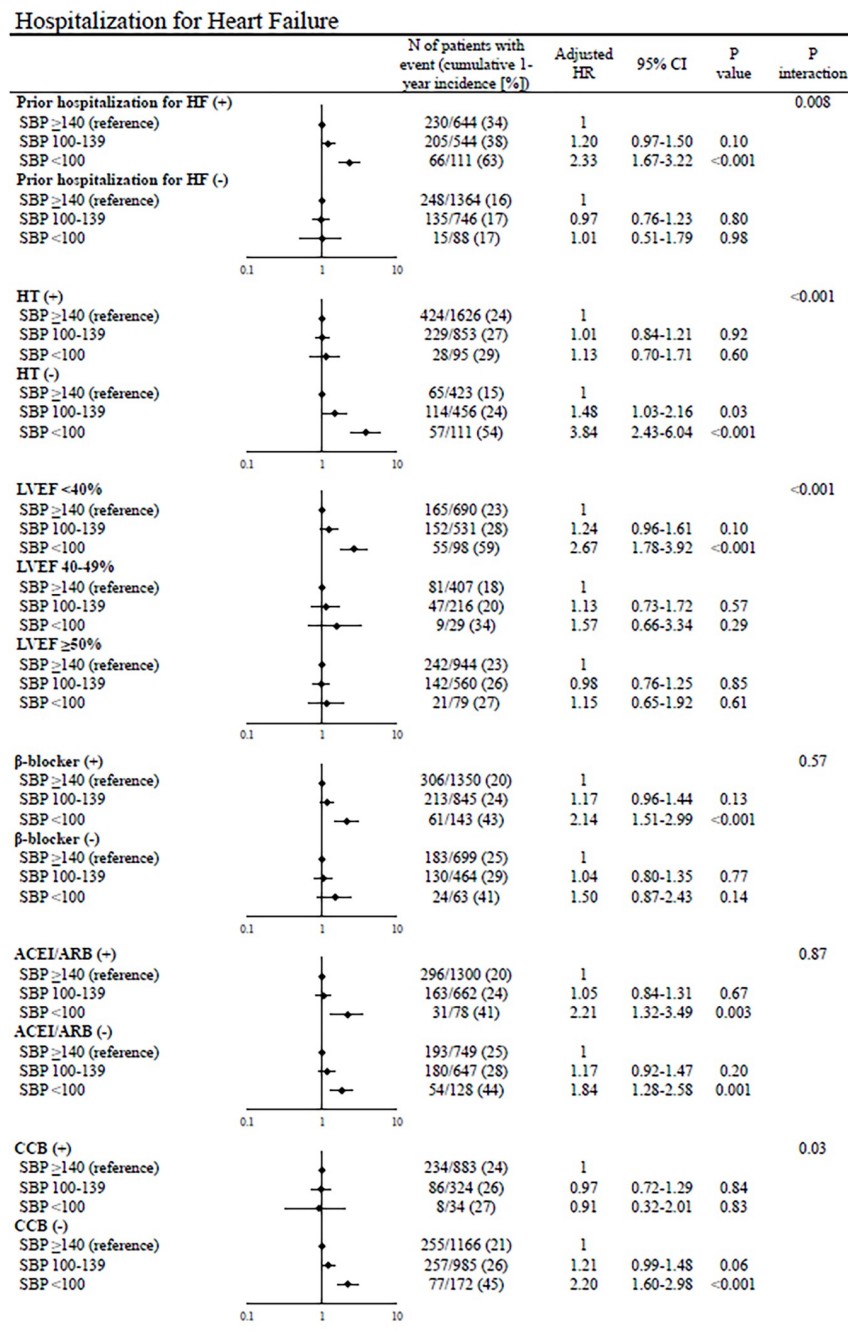

| | N of patients with event (cumulative 1-year incidence [%]) | Adjusted HR | 95% CI | P value | P interaction |
|---|---|---|---|---|---|
| **Prior hospitalization for HF (+)** | | | | | 0.008 |
| SBP ≥140 (reference) | 230/644 (34) | 1 | | | |
| SBP 100-139 | 205/544 (38) | 1.20 | 0.97-1.50 | 0.10 | |
| SBP <100 | 66/111 (63) | 2.33 | 1.67-3.22 | <0.001 | |
| **Prior hospitalization for HF (-)** | | | | | |
| SBP ≥140 (reference) | 248/1364 (16) | 1 | | | |
| SBP 100-139 | 135/746 (17) | 0.97 | 0.76-1.23 | 0.80 | |
| SBP <100 | 15/88 (17) | 1.01 | 0.51-1.79 | 0.98 | |
| **HT (+)** | | | | | <0.001 |
| SBP ≥140 (reference) | 424/1626 (24) | 1 | | | |
| SBP 100-139 | 229/853 (27) | 1.01 | 0.84-1.21 | 0.92 | |
| SBP <100 | 28/95 (29) | 1.13 | 0.70-1.71 | 0.60 | |
| **HT (-)** | | | | | |
| SBP ≥140 (reference) | 65/423 (15) | 1 | | | |
| SBP 100-139 | 114/456 (24) | 1.48 | 1.03-2.16 | 0.03 | |
| SBP <100 | 57/111 (54) | 3.84 | 2.43-6.04 | <0.001 | |
| **LVEF <40%** | | | | | <0.001 |
| SBP ≥140 (reference) | 165/690 (23) | 1 | | | |
| SBP 100-139 | 152/531 (28) | 1.24 | 0.96-1.61 | 0.10 | |
| SBP <100 | 55/98 (59) | 2.67 | 1.78-3.92 | <0.001 | |
| **LVEF 40-49%** | | | | | |
| SBP ≥140 (reference) | 81/407 (18) | 1 | | | |
| SBP 100-139 | 47/216 (20) | 1.13 | 0.73-1.72 | 0.57 | |
| SBP <100 | 9/29 (34) | 1.57 | 0.66-3.34 | 0.29 | |
| **LVEF ≥50%** | | | | | |
| SBP ≥140 (reference) | 242/944 (23) | 1 | | | |
| SBP 100-139 | 142/560 (26) | 0.98 | 0.76-1.25 | 0.85 | |
| SBP <100 | 21/79 (27) | 1.15 | 0.65-1.92 | 0.61 | |
| **β-blocker (+)** | | | | | 0.57 |
| SBP ≥140 (reference) | 306/1350 (20) | 1 | | | |
| SBP 100-139 | 213/845 (24) | 1.17 | 0.96-1.44 | 0.13 | |
| SBP <100 | 61/143 (43) | 2.14 | 1.51-2.99 | <0.001 | |
| **β-blocker (-)** | | | | | |
| SBP ≥140 (reference) | 183/699 (25) | 1 | | | |
| SBP 100-139 | 130/464 (29) | 1.04 | 0.80-1.35 | 0.77 | |
| SBP <100 | 24/63 (41) | 1.50 | 0.87-2.43 | 0.14 | |
| **ACEI/ARB (+)** | | | | | 0.87 |
| SBP ≥140 (reference) | 296/1300 (20) | 1 | | | |
| SBP 100-139 | 163/662 (24) | 1.05 | 0.84-1.31 | 0.67 | |
| SBP <100 | 31/78 (41) | 2.21 | 1.32-3.49 | 0.003 | |
| **ACEI/ARB (-)** | | | | | |
| SBP ≥140 (reference) | 193/749 (25) | 1 | | | |
| SBP 100-139 | 180/647 (28) | 1.17 | 0.92-1.47 | 0.20 | |
| SBP <100 | 54/128 (44) | 1.84 | 1.28-2.58 | 0.001 | |
| **CCB (+)** | | | | | 0.03 |
| SBP ≥140 (reference) | 234/883 (24) | 1 | | | |
| SBP 100-139 | 86/324 (26) | 0.97 | 0.72-1.29 | 0.84 | |
| SBP <100 | 8/34 (27) | 0.91 | 0.32-2.01 | 0.83 | |
| **CCB (-)** | | | | | |
| SBP ≥140 (reference) | 255/1166 (21) | 1 | | | |
| SBP 100-139 | 257/985 (26) | 1.21 | 0.99-1.48 | 0.06 | |
| SBP <100 | 77/172 (45) | 2.20 | 1.60-2.98 | <0.001 | |

**Fig 7. Subgroup analyses of the effects of admission SBP on hospitalization for heart failure.** The number of patients with events was counted through the entire follow-up period, while the cumulative incidence was truncated at 1 year. SBP = systolic blood pressure, HR = hazard ratio, CI = confidence interval, HF = heart failure, LVEF = left ventricular ejection fraction, ACEI = angiotensin-converting enzyme inhibitor, ARB = angiotensin II receptor blocker, and CCB = calcium channel blocker.

The findings from the subgroup analyses are hypothesis-generating. There was no interaction between a history of hypertension and the risk of postdischarge mortality; however, there was significant interaction between a history of hypertension and the risk of in-hospital

mortality and hospitalization for HF. Patients with high SBP may have an increased sympathetic tone, resulting in an abrupt onset of symptoms, including pulmonary congestion [12], the high rate of vasodilator use, and no increase in in-hospital mortality. In patients taking β-blocker, all-cause death occurred more frequently in the low admission SBP group than in the other two groups. The sympathetic nervous system may be more activated in ADHF patients with high admission SBP, whose prognosis can be improved by β-blocker use. Although β-blocker use can improve prognosis of HFrEF patients, patients in the low admission SBP group may suffer hypotension, which leads to increased incidence of all-cause death. β-blocker may have different class effect on the low admission SBP group. In patients taking CCB, hospitalization for HF occurred less frequently in the low admission SBP group. In patients taking CCB at discharge in the low admission SBP group, even if their blood pressure values were low on admission, it is probable that the values increased at the time of discharge and the patients were discharged after controlling blood pressure with CCB. The control of blood pressure with CCB might be effective to prevent the next surge of sympathetic tone which leads to hospitalization for HF.

The admission SBP is a simple prognostic predictor in ADHF patients because it directly reflects cardiac reserve and elasticity of vasculature. Postdischarge risk stratification using admission SBP may contribute to risk management during hospitalization and after discharge. *β*-blocker use in patients with low blood pressure on admission should be done with caution, and CCB should be added when blood pressure increased by the time of discharge, even if blood pressure on admission is low.

## Study limitations

This study has four major limitations. First, due to its observational study design, the mechanistic link between low admission SBP and higher risk of mortality was not determined. Subgroup analyses, especially those of antihypertensive drugs, were hypothesis-generating. Second, the method to measure admission SBP was not prespecified. We have no data on the mean values of two or more measurements versus single measurement; instead, we adopted SBP at hospital presentation. Third, we did not assess the sequential change of SBP, low SBP during hospitalization, and predischarge SBP. Finally, we could not assess the prescription status and patient adherence during the follow-up period. After the index hospitalization, patients taking β-blocker, angiotensin-converting enzyme inhibitor/angiotensin II receptor blocker, and CCB at discharge might have discontinued them, whereas patients not taking these medications at discharge might have started them anew.

## Conclusions

Admission SBP is useful for postdischarge risk stratification in ADHF patients. Its magnitude of the effect as a prognostic predictor may differ across clinical conditions of patients.

## Supporting information

**S1 Checklist. STROBE statement—checklist of items that should be included in reports of cohort studies.**
(PDF)

**S1 Fig. Kaplan-Meier curves for postdischarge clinical events based on further subdivided range of blood pressure (<100, 100–119, 120–139, 140–159, and ≥160 mmHg).** (A) All-cause death, (B) Cardiovascular death, (C) Noncardiovascular death, and (D) Hospitalization for HF. Follow-up was commenced on the day of discharge. SBP = systolic blood pressure,

HF = heart failure.
(PDF)

**S1 Table. Patient characteristics on admission.**
(PDF)

**S2 Table. In-hospital management.**
(PDF)

**S3 Table. In-hospital clinical outcomes.**
(PDF)

# Acknowledgments

The authors thank the staff of the KCHF registry and the other members of the participating centers.

# Author Contributions

**Conceptualization:** Yuichi Kawase.

**Data curation:** Hidenori Yaku.

**Formal analysis:** Yuichi Kawase, Takeshi Morimoto.

**Funding acquisition:** Takao Kato.

**Investigation:** Yuichi Kawase.

**Project administration:** Takao Kato.

**Supervision:** Takeshi Morimoto, Reo Hata, Ryosuke Murai, Takeshi Tada, Harumi Katoh, Kazushige Kadota, Erika Yamamoto, Hidenori Yaku, Yasutaka Inuzuka, Yodo Tamaki, Neiko Ozasa, Yusuke Yoshikawa, Moritake Iguchi, Kazuya Nagao, Yukihito Sato, Koichiro Kuwahara, Takeshi Kimura.

**Writing – original draft:** Yuichi Kawase.

**Writing – review & editing:** Takao Kato, Takeshi Morimoto, Reo Hata, Ryosuke Murai, Takeshi Tada, Harumi Katoh, Kazushige Kadota, Erika Yamamoto, Hidenori Yaku, Yasutaka Inuzuka, Yodo Tamaki, Neiko Ozasa, Yusuke Yoshikawa, Moritake Iguchi, Kazuya Nagao, Yukihito Sato, Koichiro Kuwahara, Takeshi Kimura.

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
