## [Decision Letter · Decision Letter 0]

29 Apr 2021

PONE-D-21-09482

Admission Systolic Blood Pressure as a Prognostic Predictor of Acute Decompensated Heart Failure: A Report From the KCHF Registry

PLOS ONE

Dear Dr. Kato,

Thank you for submitting your manuscript to PLOS ONE. After careful consideration, we feel that it has merit but does not fully meet PLOS ONE’s publication criteria as it currently stands. Therefore, we invite you to submit a revised version of the manuscript that addresses the points raised during the review process.

The manuscript has been carefully evaluated by 2 external reviewers and they found the manuscript potentially of interest. However, the referees have identified some conceptual and methodological problems and they have required additional information and clarifications from the authors that need to be provided.

We look forward to receiving your revised manuscript.

Kind regards,

Claudio Passino, MD

Academic Editor

PLOS ONE

Journal Requirements:

 Please provide additional details regarding participant consent. In the ethics statement in the Methods and online submission information, please ensure that you have specified (1) whether consent was informed and (2) what type you obtained (for instance, written or verbal, and if verbal, how it was documented and witnessed). If your study included minors, state whether you obtained consent from parents or guardians. If the need for consent was waived by the ethics committee, please include this information.

We note that you have indicated that data from this study are available upon request. PLOS only allows data to be available upon request if there are legal or ethical restrictions on sharing data publicly. For information on unacceptable data access restrictions, please see http://journals.plos.org/plosone/s/data-availability#loc-unacceptable-data-access-restrictions.

3a) If there are ethical or legal restrictions on sharing a de-identified data set, please explain them in detail (e.g., data contain potentially identifying or sensitive patient information) and who has imposed them (e.g., an ethics committee). Please also provide contact information for a data access committee, ethics committee, or other institutional body to which data requests may be sent.

3b) If there are no restrictions, please upload the minimal anonymized data set necessary to replicate your study findings as either Supporting Information files or to a stable, public repository and provide us with the relevant URLs, DOIs, or accession numbers. Please see http://www.bmj.com/content/340/bmj.c181.long for guidelines on how to de-identify and prepare clinical data for publication. For a list of acceptable repositories, please see http://journals.plos.org/plosone/s/data-availability#loc-recommended-repositories.

Thank you for stating the following financial disclosure:

4a)         Please clarify the sources of funding (financial or material support) for your study. List the grants or organizations that supported your study, including funding received from your institution.

4b)         State what role the funders took in the study. If the funders had no role in your study, please state: “The funders had no role in study design, data collection and analysis, decision to publish, or preparation of the manuscript.”

4c)          If any authors received a salary from any of your funders, please state which authors and which funders.

4d)         If you did not receive any funding for this study, please state: “The authors received no specific funding for this work.”

Reviewers' comments:

Reviewer's Responses to Questions

**Comments to the Author**

1. Is the manuscript technically sound, and do the data support the conclusions?

Reviewer #1: Yes

Reviewer #2: Yes

2. Has the statistical analysis been performed appropriately and rigorously? 

Reviewer #1: Yes

Reviewer #2: Yes

3. Have the authors made all data underlying the findings in their manuscript fully available?

Reviewer #1: Yes

Reviewer #2: Yes

4. Is the manuscript presented in an intelligible fashion and written in standard English?

Reviewer #1: Yes

Reviewer #2: Yes

5. Review Comments to the Author

Reviewer #1: The paper of Kato and coll is aimed to evaluate the prognostic role of systolic arterial pressure in acute decompensated heart failure.

Some points should be considered by the authors:

- In the manuscript, it is evaluated the impact of a low systolic arterial pressure on long-term prognosis. However, the role of systolic arterial pressure at the admission is well known. Although this study evaluated the long term prognosis, this is the main limitation of the study.

- The association between low systolic pressure and worse outcome was greater in patients in beta-blockers. This is not in line with previous studies and it should be better discussedn.

- Analogously, it should be clarified the relationship between greater hospitalization for heart failure in patients with CCB.

- Is there any relationship between changes in arterial pressure at admission and pre-discharge and prognosis? This could be an analysis providing data useful for daily clinical practice

- The relationship among arterial systolic pressure and HFrEF, HFpEF and HFmrEF should be discussed

- Authors included in the multivariate model 19 clinically relevant variables. The method used to select these variables should be described.

Reviewer #2: In the present paper, Kawase and colleagues aimed to explore the prognostic value of admission Systolic Blood Pressure (SBP) in Acute Decompensate Heart Failure (ADHF) patients. The study is a prospective, observational, multicenter cohort study involving 19 hospitals in Japan.

The paper is rather focused and provides further insight on the value of admission SBP in the prognosis of ADHF. The large number of patients enrolled in the study is undoubtedly an element of strength. Furthermore the statistical analysis is well described and performed. Still, the paper sufferes from some major methodological limitations, as listed below.

Major points

- Lee and colleagues (Circulation: Heart Failure, 2009) found that the initial blood pressure at the time of acute HF presentation was only weakly associated with discharge blood pressure. Why did you choose the admission SBP and not the discharge SBP to explore the post-discharge clinical outcomes?

- In the “Discussion”, you did not explain if the findings could have clinical implications or suggestions on how to manage HF therapies. Please, add a paragraph in the “Discussion”. It would strengthen the value of the whole paper.

- The whole in-hospital outcomes section seems to be not included neither in primary outcomes nor in secondary ones, why? Please motivate your choice if not modifiable.

- The three groups have wide range of SBP values. More narrow limits between different groups would have provided a more accurate statistical analysis. What is your opinion and why did you choose those threshold?

Minor points

- Do you have any data of compliance to the therapy for the three different groups of admission SBP?

- Some syntax errors are present in the paper, please revise the global English form

6. PLOS authors have the option to publish the peer review history of their article (what does this mean?). If published, this will include your full peer review and any attached files.

Reviewer #1: No

Reviewer #2: No

---

## [Author Response · Author response to Decision Letter 0]

26 May 2021

Respose to Reviewer 1:

Thank you for your review of our paper. We have answered each of your points below.

Reviewer #1: The paper of Kato and coll is aimed to evaluate the prognostic role of systolic arterial pressure in acute decompensated heart failure.

Some points should be considered by the authors:

- In the manuscript, it is evaluated the impact of a low systolic arterial pressure on long-term prognosis. However, the role of systolic arterial pressure at the admission is well known. Although this study evaluated the long term prognosis, this is the main limitation of the study.

We agree with you and have incorporated this suggestion into our paper. In this study, we examined whether the validity of low admission SBP in predicting clinical outcomes is consistent across various clinical subtypes of ADHF patients, and whether the impact of admission SBP on long-term prognosis tended to be similar to that reported by previous studies in modern medical care for ADHF.

We have rewritten the text (page 5, lines 60-64) as “The aim of this study was to determine whether the impact of admission SBP on long-term prognosis in modern medical care for ADHF tended to be similar to that reported by previous studies by exploring the prognostic value of low admission SBP using the data from a large Japanese observational database of ADHF patients.”.

- The association between low systolic pressure and worse outcome was greater in patients in beta-blockers. This is not in line with previous studies and it should be better discussed.

We agree with your assessment. We added the following sentence to the text (page 31, lines 329-335).

“In the patients with β-blocker, all-cause death occurred more frequently in the low admission SBP group, and that occurred less frequently in other groups. Sympathetic nervous system may be more activated in ADHF patients with high admission SBP. It may be possible to improve the prognosis by β-blocker in those patients. In contrast, β-blockers are drugs that improve prognosis in HFrEF patients, but in the low admission SBP group, hypotension as an adverse event may be more likely to occur, which may lead to an increase in the incidence of all-cause death. Otherwise, B-blocker may have different class effect in the low admission SBP group.”

- Analogously, it should be clarified the relationship between greater hospitalization for heart failure in patients with CCB.

We agree with your assessment. We added the following sentence to the text (page 31-32, lines 335-341).

“In patients taking CCB, hospitalization for HF occurred less frequently in the low admission SBP group. In patients taking CCB at discharge in the low admission SBP group, even if their blood pressure values were low on admission, it is probable that the values increased at the time of discharge and the patients were discharged after controlling blood pressure with CCB. The control of blood pressure with CCB might be effective to prevent the next surge of sympathetic tone which leads to hospitalization for HF.”

- Is there any relationship between changes in arterial pressure at admission and pre-discharge and prognosis? This could be an analysis providing data useful for daily clinical practice

You have raised an important point; however, we explored the prognostic value of admission systolic blood pressure in ADHF patients in this study. Blood pressure is determined by cardiac output and systemic resistance. In ADHF settings, the prompt adaptation of cardiac output and elasticity of arteries and vascular bed was decompensated. Therefore, we could consider a few plausible mechanisms explaining the inverse association between admission SBP and poor prognosis of HF patients. We could not assessed changes in arterial pressure at admission and pre-discharge and prognosis.

We added the following sentence to Study limitaions (page 33, lines 355-356).

“Third, we did not assess the sequential change of SBP, low SBP during hospitalization, and predischarge SBP.”

- The relationship among arterial systolic pressure and HFrEF, HFpEF and HFmrEF should be discussed

We agree with your assessment. We added the following sentence to the text (page 30-31, lines 306-317).

“There are a few plausible mechanisms of the inverse association between admission SBP and poor prognosis of HF patients. Blood pressure is determined by cardiac output and systemic resistance. In ADHF settings, the prompt adaptation of cardiac output and elasticity of arteries and vascular bed was decompensated. In HFrEF patients, low cardiac output was related to low admission SBP; thus, low admission SBP was related to both high in-hospital mortality and poor postdischarge outcomes. In contrast, when cardiac output is normal or slightly reduced, a hypertensive response is expected, particularly in hypertensive patients, as a result of sympathetic and neurohormonal activation. Thus, at the time of discharge when they were under drug therapy, admission SBP may have had smaller effects on HFmrEF and HFpEF patients. In this study, we examined whether admission SBP was a predictor of long-term prognosis, assuming that the response of blood pressure during acute exacerbation of HF was due to the mechanism described above.”

- Authors included in the multivariate model 19 clinically relevant variables. The method used to select these variables should be described.

Thank you for pointing out. We selected them based on the clinical relevance to prognosis and the mean values of the data to ensure consistency with our previous report. We structured the listing in “demographical-HF related-comorbidities-living status-admission vital signs-admission lab values-discharge medications” for clarity.

We have rewritten the text (page 9-10, lines 138-148) as “We included the following 19 clinically relevant risk-adjusting variables into the model: demographical variables (age ≥80 years, sex, and body mass index <22 kg/m2), variables related to heart failure (prior hospitalization for HF, LVEF <40% by echocardiography), variables related to comorbidities (atrial fibrillation or flutter, hypertension, diabetes mellitus, prior myocardial infarction, prior stroke, current smoker, and chronic lung disease), living status (living alone and ambulatory), vital signs at presentation (admission heart rate <60 bpm), laboratory tests on admission (estimated glomerular filtration rate <30 mL/min/1.73 m2, albumin <3.0 g/dL, sodium <135 mmol/L, and anemia) as well as the three groups based on admission SBP (S1 Table). We selected them based on the clinical relevance to prognosis and the mean values of the data to ensure consistency with our previous report”.

We have rewritten the text (page 10-11, lines 158-168) as “We included the following 21 clinically relevant risk-adjusting variables into the model: demographical variables (age ≥80 years, sex, and body mass index <22 kg/m2), variables related to heart failure (prior hospitalization for HF, LVEF <40% by echocardiography), variables related to comorbidities (atrial fibrillation or flutter, hypertension, diabetes mellitus, prior myocardial infarction, prior stroke, current smoker, and chronic lung disease), living status (living alone and ambulatory), laboratory tests on admission (estimated glomerular filtration rate <30 mL/min/1.73 m2, albumin <3.0 g/dL, sodium <135 mmol/L, and anemia), and medications at discharge (angiotensin converting enzyme inhibitors or angiotensin II receptor blockers, and β-blockers) as well as the three groups based on admission SBP. We selected them on the basis of the clinical relevance to prognosis and the mean values of the data to ensure consistency with our previous report”.

Respose to Reviewer 2:

Thank you for your review of our paper. We have answered each of your points below.

Reviewer #2: In the present paper, Kawase and colleagues aimed to explore the prognostic value of admission Systolic Blood Pressure (SBP) in Acute Decompensate Heart Failure (ADHF) patients. The study is a prospective, observational, multicenter cohort study involving 19 hospitals in Japan.

The paper is rather focused and provides further insight on the value of admission SBP in the prognosis of ADHF. The large number of patients enrolled in the study is undoubtedly an element of strength. Furthermore the statistical analysis is well described and performed. Still, the paper sufferes from some major methodological limitations, as listed below.

Major points

- Lee and colleagues (Circulation: Heart Failure, 2009) found that the initial blood pressure at the time of acute HF presentation was only weakly associated with discharge blood pressure. Why did you choose the admission SBP and not the discharge SBP to explore the post-discharge clinical outcomes?

You have raised an important point; however, we explored the prognostic value of admission systolic blood pressure in ADHF patients in this study. Blood pressure is determined by cardiac output and systemic resistance. In ADHF settings, the prompt adaptation of cardiac output and elasticity of arteries and vascular bed was decompensated. Therefore, we could consider a few plausible mechanisms explaining the inverse association between admission SBP and poor prognosis of HF patients. We could not assessed changes in arterial pressure at admission and pre-discharge and prognosis.

We added the following sentence to Study limitaions (page 33, lines 355-356).

“Third, we did not assess the sequential change of SBP, low SBP during hospitalization, and predischarge SBP.”

- In the “Discussion”, you did not explain if the findings could have clinical implications or suggestions on how to manage HF therapies. Please, add a paragraph in the “Discussion”. It would strengthen the value of the whole paper.

Thank you for pointing out. We added the following sentence to the text (page 32, lines 343-347).

“Postdischarge risk stratification using admission SBP may contribute to risk management during hospitalization and after discharge. β-blocker use in patients with low blood pressure on admission should be done with caution, and CCB should be added when blood pressure increased by the time of discharge, even if blood pressure on admission is low.”

- The whole in-hospital outcomes section seems to be not included neither in primary outcomes nor in secondary ones, why? Please motivate your choice if not modifiable.

We agree with you and have incorporated this suggestion throughout the paper. We added in-hospital mortalities to the secondary outcomes.

We have rewritten the text (page 8, lines 108-111) as “The secondary outcome measures included in-hospital all-cause death, in-hospital cardiovascular death, in-hospital noncardiovascular death, cardiovascular death after discharge, noncardiovascular death after discharge, and hospitalization for HF.”.

We have rewritten the text (page 8-9, lines 124-129) as “Subgroup analyses of the association of admission SBP with primary and secondary outcome measures during hospitalization and after discharge were conducted for prior hospitalization for HF, hypertension, and LVEF. Also, those after discharge alone were conducted for β-blocker use, angiotensin converting enzyme inhibitor or angiotensin II receptor blocker use, and calcium channel blocker (CCB) use at discharge”.

We added the following sentence to the text (page 13, lines 207-211).

“The rate of ventricular tachycardia or fibrillation was significantly higher in the low admission SBP group than in the intermediate and high admission SBP groups (11%, 5%, and 3.6%, respectively) (S3 Table). The rate of worsening renal function was significantly lower in the low admission SBP group than in the intermediate and high admission SBP groups (19%, 29%, and 40%, respectively) (S3 Table).”

We have rewritten the text (page 29, lines 286-289) as “In the entire cohort, lower admission SBP was associated with higher risk of in-hospital and postdischarge all-cause and cardiovascular death and hospitalization for HF, but not with in-hospital and postdischarge noncardiovascular death”.

In addition, Table 1 and Fig 2 have been added.

- The three groups have wide range of SBP values. More narrow limits between different groups would have provided a more accurate statistical analysis. What is your opinion and why did you choose those threshold?

Thank you for pointing out. In this study, we divided into three categories according to the clinical scenario classification. We added the sensitivity analysis in further subdivided blood pressure categories (<100, 100-119, 120-139, 140-159, ≥160).

We added the following sentence to the text (page 29-30, lines 293-298).

In addition, S4 Fig have been added.

Minor points

- Do you have any data of compliance to the therapy for the three different groups of admission SBP?

You have raised an important point; however, we could not assessed compliance to the medication therapy.

We added the following sentence to Study limitaions (page 33, lines 356-360).

“Finally, we could not assess the prescription status and patient adherence during the follow-up period. After the index hospitalization, patients taking β-blocker, angiotensin-converting enzyme inhibitor/angiotensin II receptor blocker, and CCB at discharge might have discontinued them, whereas patients not taking these medications at discharge might have started them anew.”

- Some syntax errors are present in the paper, please revise the global English form

Thank you for pointing out. We modified it to an appropriate English form.

---

## [Decision Letter · Decision Letter 1]

18 Jun 2021

Admission systolic blood pressure as a prognostic predictor of acute decompensated heart failure: a report from the KCHF registry

PONE-D-21-09482R1

Dear Dr. Kato,

We’re pleased to inform you that your manuscript has been judged scientifically suitable for publication and will be formally accepted for publication once it meets all outstanding technical requirements.

Kind regards,

Claudio Passino, MD

Academic Editor

PLOS ONE

Additional Editor Comments (optional):

Reviewers' comments:

Reviewer's Responses to Questions

**Comments to the Author**

1. If the authors have adequately addressed your comments raised in a previous round of review and you feel that this manuscript is now acceptable for publication, you may indicate that here to bypass the “Comments to the Author” section, enter your conflict of interest statement in the “Confidential to Editor” section, and submit your "Accept" recommendation.

Reviewer #1: All comments have been addressed

Reviewer #2: All comments have been addressed

2. Is the manuscript technically sound, and do the data support the conclusions?

Reviewer #1: Partly

Reviewer #2: Yes

3. Has the statistical analysis been performed appropriately and rigorously? 

Reviewer #1: Yes

Reviewer #2: Yes

4. Have the authors made all data underlying the findings in their manuscript fully available?

Reviewer #1: Yes

Reviewer #2: Yes

5. Is the manuscript presented in an intelligible fashion and written in standard English?

Reviewer #1: Yes

Reviewer #2: Yes

6. Review Comments to the Author

Reviewer #1: The authors answered to the comments raised. The paper has been improved, although its contents substantially replicate data already published.

Reviewer #2: Authors responded satisfactorily to all comments and now the paper is improved and worth of pubblication. In particular, statistical analyses and "results" section have been re-organized and better elucidated. In "Discussion" section authors added further comments on the clinical implications that this paper coud have on daily practice. English form has been revised properly.

7. PLOS authors have the option to publish the peer review history of their article (what does this mean?). If published, this will include your full peer review and any attached files.

Reviewer #1: No

Reviewer #2: **Yes: **Alessandro Valleggi

---

## [Editor Report · Acceptance letter]

24 Jun 2021

PONE-D-21-09482R1 

Admission systolic blood pressure as a prognostic predictor of acute decompensated heart failure: a report from the KCHF registry 

Dear Dr. Kato:

I'm pleased to inform you that your manuscript has been deemed suitable for publication in PLOS ONE. Congratulations! Your manuscript is now with our production department. 

Kind regards, 

on behalf of

Prof. Claudio Passino 

Academic Editor

PLOS ONE